# Feature-Oriented CBCT Self-Calibration Parameter Estimator for Arbitrary Trajectories: FORCAST-EST

**Christian Tönnes** [1,2,*] and **Frank G. Zöllner** [1,2]

1   Computer Assisted Clinical Medicine, Medical Faculty Mannheim, Heidelberg University, Theodor-Kutzer-Ufer 1, 68159 Mannheim, Germany
2   Mannheim Institute for Intelligent Systems in Medicine, Medical Faculty Mannheim, Heidelberg University, Theodor-Kutzer-Ufer 1, 68159 Mannheim, Germany
*   Correspondence: christian.toennes@medma.uni-heidelberg.de

**Abstract:** Background: For the reconstruction of Cone-Beam CT volumes, the exact position of each projection is needed; however, in some situations, this information is missing. Purpose: The development of a self-calibration algorithm for arbitrary CBCT trajectories that does not need initial positions. Methods: Projections are simulated in a spherical grid around the center of rotation. Through using feature detection and matching, an acquired projection is compared to each simulated image in this grid. The position with the most matched features was used as a starting point for a fine calibration with a state-of-the-art algorithm. Evaluation: This approach is compared with the calibration of nearly correct starting positions when using FORCASTER and CMA-ES minimization with a normalized gradient information (NGI) objective function. The comparison metrics were the normalized root mean squared error, structural similarity index, and the dice coefficient, which were evaluated on the segmentation of a metal object. Results: The parameter estimation for a regular Cone-Beam CT with a 496 projection took 1:26 h with the following metric values: NRMSE = 0.0669; SSIM = 0.992; NGI = 0.75; and Dice = 0.96. FORCASTER with parameter estimation took 3:28 h with the following metrics: NRMSE = 0.0190; SSIM = 0.999; NGI = 0.92; and Dice = 0.99. CMA-ES with parameter estimation took 5:39 h with the following metrics: NRMSE = 0.0037; SSIM = 1.0; NGI = 0.98; and Dice = 1.0. Conclusions: The proposed algorithm can determine the parameters of the projection orientations for arbitrary trajectories with enough accuracy to reconstruct a 3D volume with low errors.

**Keywords:** CBCT; cone-beam CT; computed tomography; calibration

## 1. Introduction

To reconstruct a Cone-Beam Computed Tomography image, several X-ray images are required and must be input into a reconstruction algorithm. This algorithm must know at which position and orientation each X-ray image was made. Several methods have been developed to solve this problem, and they come in two broad categories, offline and online calibration.

Offline calibration uses phantoms with markers, which are often small metal beads that are imaged for each projection where the position and orientation can be calculated [1–7]. The phantom is imaged with a trajectory, and the correction factors are calculated. Then, these correction factors are used when reconstructing the images from this trajectory. This requires a dedicated run of the trajectory with the phantom.

Online calibration uses the acquired projections and then uses prior information about the imaged object [8–11], or it minimizes a cost function that is defined on the reconstructed image [12–15]. A review of current approaches was published by Hatamikia et al. [16]. This approach does not need a dedicated phantom nor an extra run of the trajectory for the calibration, but it does have other constraints. Some use a prior image on which the

projections are registered when using 2D–3D registration algorithms, others require the trajectory to have a specific, often circular, shape.

Using a 2D–3D registration of the acquired projections on a prior image allows for the calibration of fully arbitrary trajectories [17], and the registration is faster if the initial parameters are close to the actual parameters. The position reported by the CBCT system is usually close enough for a quick and good calibration with state-of-the-art algorithms. But, if this information is not available (e.g., portable C-Arms, continuous acquisition/fluoroscopy mode of the Artis Zeego), other means of obtaining these initial parameters are required.

One option is to track the C-Arm externally with inertia sensors [18,19]. These sensors can be attached to the C-Arm and, after calibrating the sensors, they track the inertia in all three dimensions. With this inertia data and a known starting position, the movement of the C-Arm can be calculated.

Also possible is the use of 3D cameras [20,21]. Here, the position of the C-Arm is observed by tracking optical markers through using either a camera attached to the X-ray source or two external cameras.

This paper presents another option that uses a prior image to simulate forward projections from different angles, and then uses feature matching to find the one that fits bests for each acquired projection. In this way, the parameters for each projection can be estimated.

## 2. Materials and Methods

### 2.1. Projection and Optimization Parameters

The algorithm uses an intrinsic coordinate system that is bound to the detector plane. The three vectors to define the position and orientation of the acquired image are shown in Figure 1. The vector $\vec{d}$ points from the middle of the detector to the source, and the length is the distance between the detector and the source. The vectors $\vec{u}$ and $\vec{v}$ describe the direction and spacing of the detector elements, respectively, and they point from the center of one detector element to the center of the left/top neighbor. The size of each vector is the spacing between elements.

The three unit vectors $\vec{x}$, $\vec{y}$, and $\vec{z}$ that make up the coordinate system are parallel to the vectors $\vec{u}$, $\vec{v}$ and $\vec{d}$, respectively, and the point of origin is the isocenter, which is the center of the CT image.

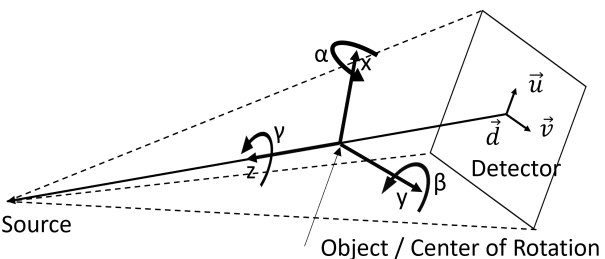

**Figure 1.** Overview of the coordinate system, parameters, and the degrees of freedom [10].

These vectors are also used for all movements and rotations. There are three rotations, one around each of the vectors $x$, $y$, and $z$, and three translations that are also along these vectors.

### 2.2. Feature Points Matching

The algorithm depends on feature points; these are found within each image through using the AKAZE [22] algorithm. The parameters used for AKAZE were as follows—threshold: 0.0005, four Octaves, and five Octave Layers. AKAZE also generates a description vector for each feature point, and these descriptors can be used to compare to points through using the Hamming distance. To find the matching features between images, the Hamming distances between all feature descriptors of one image to all descriptors from the other image are calculated. Then, for every feature in the calibration image, the features

with the lowest Hamming distance $d_1$, and the one with the second lowest distance $d_2$ are selected. On these two distances, Lowe's ratio test [23] is applied. This compares the distance with a ratio $r$, and this is performed in order to check that the smaller distance is much smaller than the second best match ($d_1 < r * d_2$). When the test succeeds, the feature point with the smaller distance is used to form a matching pair of feature points. If the distances do not comply with this test, no matching pair was found.

Afterward, all found pairs are filtered by finding the ones that match to the same feature point and those are removed.

The last step involves discarding pairs where the Euclidean distance between the points in the matched pairs is more than one standard deviation from the mean distance of all pairs.

*2.3. Algorithm*

An overview of the estimation process is in Figure 2, and this will be explained in more detail in the following section.

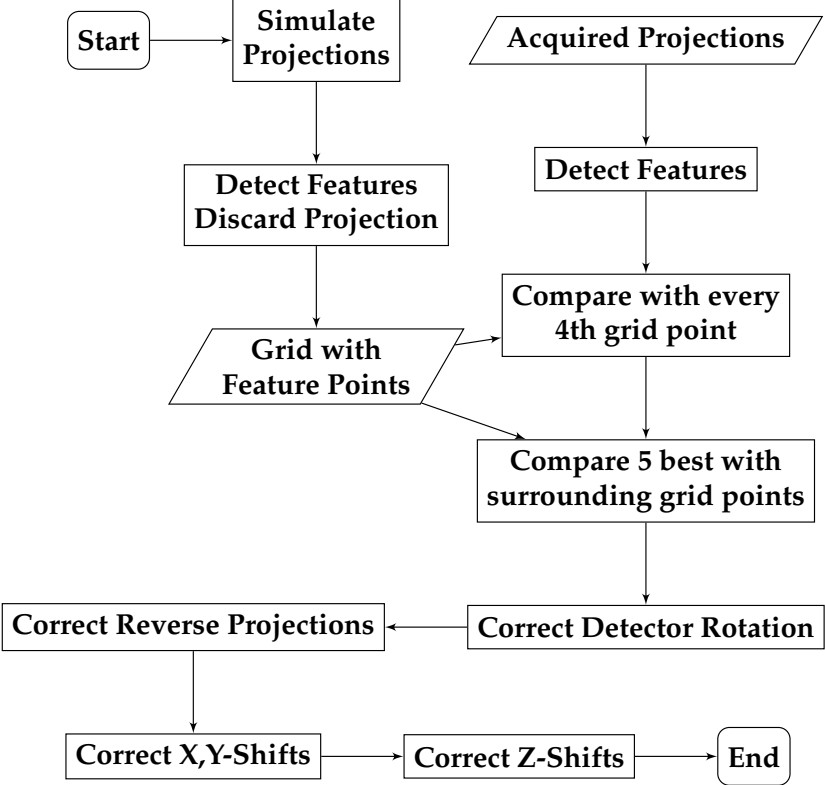

**Figure 2.** Pipeline of the estimation algorithm. Rectangles are the processes, and the trapezoids are the store data.

The algorithm is initiated with the acquired images, a prior CT, and geometry information about the size of the detector, as well as the distance to the source and to the isocenter. The first step of the algorithm is to generate simulated projections in a regular grid around the center of the prior CT image. The grid has 95 points each for the rotations around the x- and y-axis, which results in a grid with a spacing of 3.8°. Further, three different detector rotations are used, 0°, 120°, and 240°. On each of these images, the AKAZE algorithm detects features and extracts the feature descriptors. The simulated projections are then deleted, and only the feature points, descriptors, and projection rotations ($\alpha$, $\beta$, and $\gamma$) are saved. They are calculated once and then used for all further calibration.

To find the approximate position of an acquired image, the algorithm first detects the features. They are then matched with each set of features from the simulated grid



projections and the matched feature points are counted. To save time, the algorithm operates along this grid with a step size of four, and it then selects the five grid points with the most matched feature points. Next, the grid points surrounding these five points are compared to the acquired image in the same way. For the grid point that has the most matched pairs, the projection rotations are returned. These are the current approximation of the rotations $\hat{\alpha}_1$, $\hat{\beta}_1$, and $\hat{\gamma}_1$ (Figure 3a).

This selected grid position is most likely the closest to the target position, but it can also be a projection from the opposite side. These projections, simulated from the wrong side, will be corrected later.

Before that, the detector rotation is approximated. The average value of the feature point coordinates is used as the center point; this is performed separately for the real and simulated image. Then, the coordinates of the feature points are converted to polar coordinates by using the averaged center point as the zero point. The angle for each feature point in the simulated image is subtracted from the angle of the matching feature point in the real image. The median of these differences is the new approximated detector rotation $\hat{\gamma}_2$. Listing 1 shows this in pseudocode.

---

**Listing 1.** Approximation function for the detector rotation.

---

```
1   def approximate_detector_rotation(current_parameters):
2       # simulate projection and track features
3       simulated_projection = ForwardProjection(current_parameters)
4       simulated_feature_points = trackFeatures(simultaded_projection,
        ↪  real_projection)
5       # calculate center point
6       simulated_mid = mean(real_feature_points, axis=0)
7       real_mid = mean(simulated_feature_points, axis=0)
8       sim_points = simulated_feature_points - simulated_mid
9       real_points = real_feature_points - real_mid
10      # calculate angle
11      angles = (arctan2(sim_points[:,0], sim_points[:,1])
12              -arctan2(real_points[:,0], real_points[:,1])) * 180.0/PI
13      angles[angle<-180] += 360
14      angles[angle>180] -= 360
15      detector_angle = median(angles)
16      # test in which direction to rotate
17      proj = ForwardProjection(applyRotation(current_parameters,
        ↪  0,0,-detector_angle))
18      points = trackFeatures(proj, real_projection)
19      diffn = points - real_feature_points
20      proj = ForwardProjection(applyRotation(current_parameters,
        ↪  0,0,+detector_angle))
21      points = trackFeatures(proj, real_projection)
22      diffp = points - real_feature_points
23      if sum( abs(diffn) ) < sum( abs(diffp) ):
24          return -detector_angle
25      else:
26          return detector_angle
```

---

A similar approach is taken to correct the projections taken from the opposite direction. Four projections are simulated with different rotation parameters, as well as the approximate rotations with a detector rotation of 180° ($\hat{\gamma}_2 + \pi$). This is conducted from the opposite side that rotates around the x-axis ($\hat{\alpha}_1 + \pi$) and around the y-axis ($\hat{\beta}_1 + \pi$). For these four projections, features are detected and matched, and the projection with the lowest mean Euclidean distance between the matched points is then used.

The result of this first step is a rough calibration. As such, in the next step, this rough calibration is further refined. The translational misalignment is corrected using the method described by Tönnes et al. [10]. The median Euclidean distance between the matching points is used to move the projection in the x and y directions. The z-translation is corrected by calculating the distance between the feature points within each image, and by then dividing the distances of one image by those of the other image results in the zoom factor. This ratio and the distance between the source and the isocenter are multiplied to give the new distance.

After correcting the translations along the x-, y- and z-axis, the previously described procedure that is used to correct the detector rotation is applied once more (Figure 3b).

The resulting parameters can then be used to run a state-of-the-art calibration algorithm and fully calibrate the trajectory (Figure 3c).

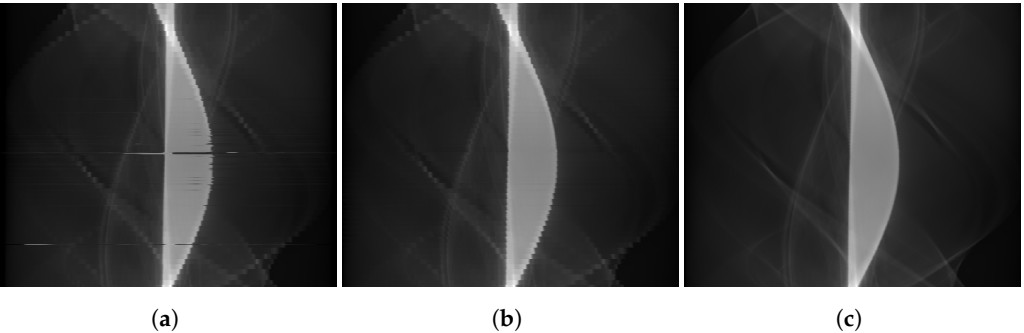

| (a) | (b) | (c) |

**Figure 3.** The sinograms for the different steps in the algorithm. From left to right is as follows: (**a**) coarse estimate, (**b**) refined estimate, and (**c**) the refined estimate with FORCASTER when using the NGI objective.

### 2.4. Image Data

In this paper, the data from Tönnes et al. [10] were used, which were obtained from a CT scan of a lumbal spine phantom with an inserted metal object. The reconstructions can be seen in Figure 4.

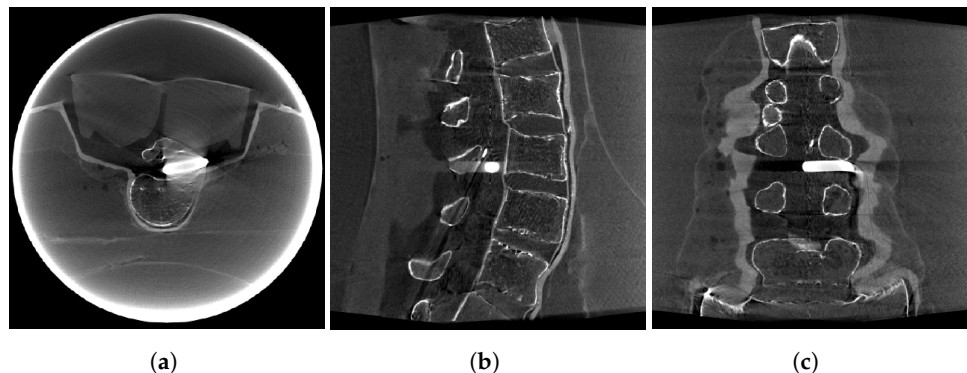

| (a) | (b) | (c) |

**Figure 4.** CBCT used as a prior image. (**a**) Transversal slice. (**b**) Sagittal slice. (**c**) Coronal slice.

Furthermore, a sinusoidal trajectory, acquired shortly after the abovementioned CT scan, is used. The phantom is not moved in between. The sinusoidal trajectory is acquired in a step-and-shoot mode, which means moving the C-Arm to each of the 161 positions on this trajectory, and then acquiring a single X-ray image with the standard protocol called "P16_DR_L" at 70 keV and with the mAs controlled by the Artis Zeego System.

The third trajectory is acquired using the continuous acquisition mode and by moving the C-Arm during acquisition. This one has the problem that was mentioned in the introduction in that it does not contain positional information for the individual frames. This trajectory is an arc around the object tilted by 28°, with 70 keV and 30 frames per s. The exposure time and tube current are managed by the Artis Zeego System; furthermore,

the average pulse width is 3.5 ms, with an average current of 35 mA. It consists of 666 individual projections.

All three sinograms are shown in Figure 5. Since the sinograms are three-dimensional, only two slices are shown, and each of them is cut through the center of all individual projections, both horizontally and vertically.

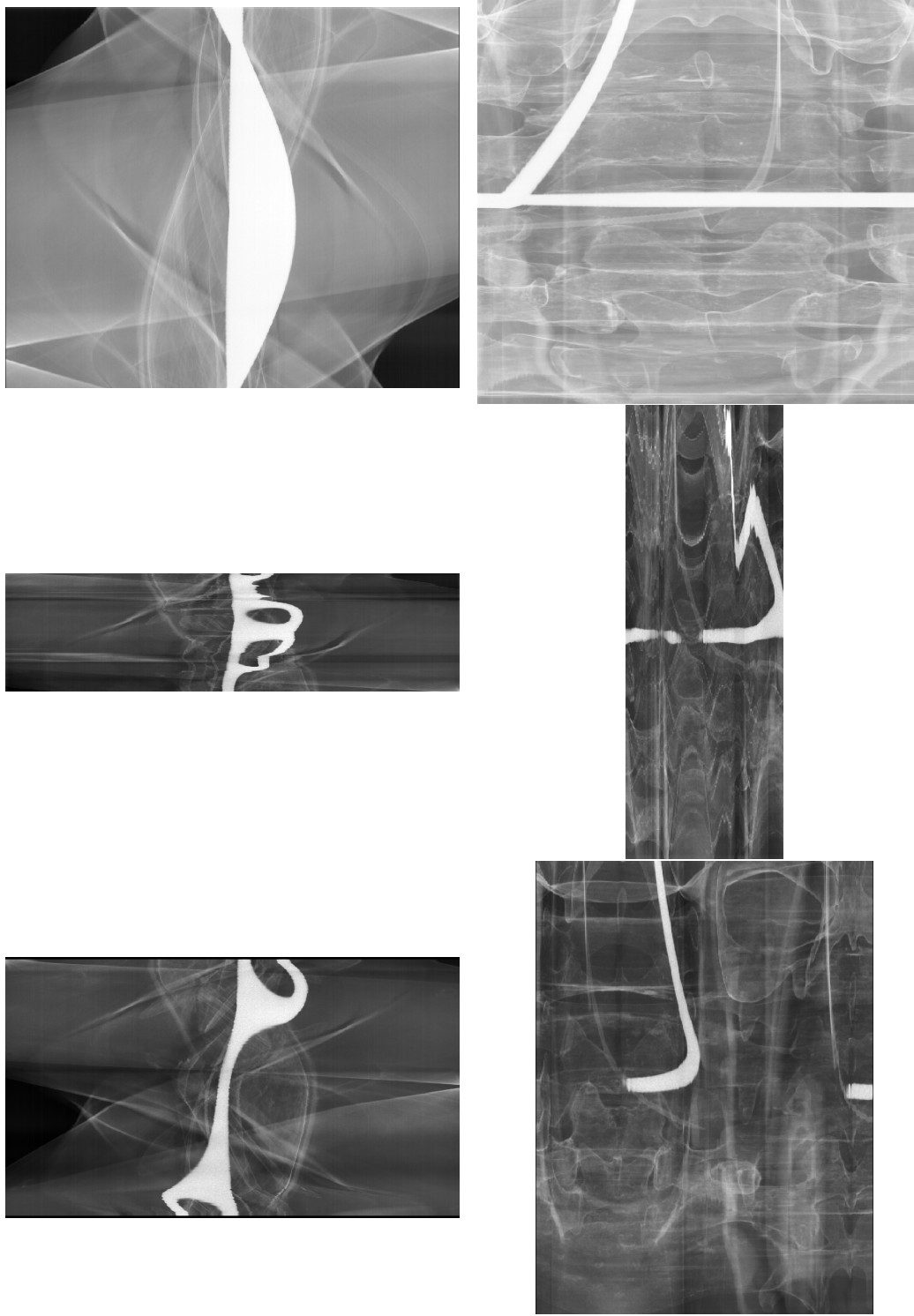

**Figure 5.** Two slices through the acquired three-dimensional sinograms. The top row is the standard CBCT, the middle row is the sinusoidal trajectory, and the bottom row is a continuous acquisition of a circular arc of 400°.

*2.5. Evaluation*

To evaluate the quality of the estimator, the estimated parameters were used as inputs for the FORCASTER [10] algorithm and the algorithm by Oudah et al. [17] (which uses a CMA-ES minimizer with the normalized gradient information (NGI) as the objective function).

The calibrated trajectories are then reconstructed with the FDK algorithm, which is part of the astra toolbox [24]. The images become cropped to the field of view, and no further post-processing is performed. The calibrated parameters are also used to generate a forward projection using the prior image; this simulated sinogram is compared to the simulated forward projections of the state-of-the-art calibration algorithm. The continuous acquisition sinogram is compared to the acquired data since there are no correctly calibrated parameters.

2.5.1. Metrics

The structural similarity index (SSIM) [25] (Equation (1)) is the normalized gradient information (NGI) [26], and the normalized root mean squared error (NRMSE) (Equation (2)) are evaluated on the projections that are simulated with the parameters from the calibrated trajectory in comparison to the forward projections of the reference calibration.

$$SSIM(x,y) = \frac{2\mu_x\mu_y(2\sigma_{xy} + c_2)}{(\mu_x^2 + \mu_y^2 + c_1)(\sigma_x^2 + \sigma_y^2 + c_2)} \tag{1}$$

$$NRMSE(x,y) = \frac{RMSE(x,y)}{||x||_F} \tag{2}$$

$$RMSE(x,y) = \sqrt{\frac{1}{N}\sum_i^N (x_i - y_i)^2} \tag{3}$$

In this equation, $\mu_x$ is the mean value of $x$; $\sigma_x$ the standard deviation; $c_1$ and $c_2$ are the constants; $N$ : This is changed to be italics format to keep consistent with the equation, please confirm. Same below. is the number of voxels; and $||x||_F$ is the Frobenius norm of $x$.

The reconstructions obtained after calibration and cropping to the point of view are also compared through using the same metrics. Additionally, the vertical part of the large metal object in the center of the phantom is segmented, and the Dice coefficient on the segmentations is then calculated. All metrics are applied to each 2D slice of the images and then averaged.

2.5.2. System Specifications

The algorithms were run on a system with an AMD Ryzen 9 7900X CPU, 128 GB RAM, and NVIDIA GeForce RTX 2070 SUPER. Due to each projection being independent of the others, the algorithm can be easily parallelized. In this paper, ten parallel processes were used. Python version 3.9.9 the following packages was used: astra-toolbox 2.0 [24], scipy 1.7.3, skimage 0.19, numpy 1.21.4, and opencv 4.5.4.

## 3. Results

The results obtained from comparing the sinograms are in Table 1. The similarity index and NGI are low for the coarse estimate, but they increased significantly after refining. When using the FORCASTER algorithm with the estimated positions, the SSIM and NRMSE were comparable to the ones reported by Tönnes et al. [10].

Similar to the results from the sinograms, the metrics evaluated on the reconstructions, shown in Table 2, showed a significant improvement in the refined estimate over the coarse one. Here, the NRMSE and Dice functions, after estimating and applying FORCASTER, were also comparable to the ones previously reported. There was no significant difference ($p = 0.56$) between the Dice values of the two reconstructions when using the state-of-the-art

algorithms and refined estimates; however, the other metrics—NGI, SSIM, and NRMSE—were significantly better for the CMA-ES calibration.

The reconstructed images are shown in Figure 6. The top row contains the reconstruction after only performing the rough estimate. There were obvious artifacts visible, which can be seen in the left column. These artifacts are regularly spaced and rotated around the center in the plane of the acquisition trajectory. Several edges were reconstructed twice with a slight offset. In the second row, the refined estimate was used for reconstruction. The radial artifacts are still there, but the double edges are now consolidated. Finally, the reconstruction without these radial artifacts is in the bottom row.

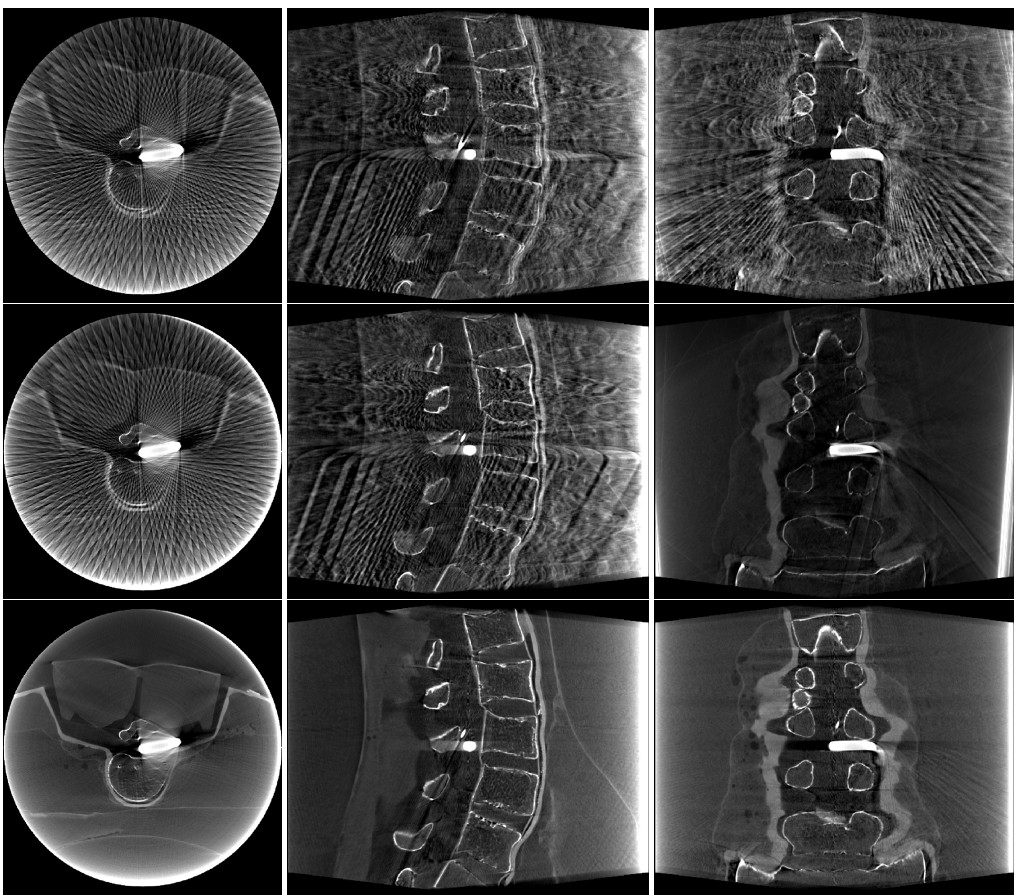

**Figure 6.** Reconstructions of the different steps of the algorithms. From top to bottom is as follows: Coarse estimate, Refined estimate and Refined estimate + CMA-ES when using the NGI objective.

**Table 1.** Results for the metrics evaluation on the CBCT sinogram. The two bottom rows used the refined estimate as the input for the FORCASTER algorithm, and this was achieved once with the NGI objective and once with the objective when based on feature points. The values marked with an * had significant ($p < 0.05$) changes compared to the refined estimate.

| Algorithm | Runtime [hh:mm] | NGI | SSIM | NRMSE |
|---|---|---|---|---|
| Coarse Estimate | 01:06 | 0.397 * | 0.944 * | 0.2327 * |
| Refined Estimate | 01:26 | 0.753 | 0.992 | 0.0669 |
| Est. + FORCASTER | 03:28 | 0.919 * | 0.999 * | 0.0190 * |
| Est. + CMA-ES & NGI | 05:39 | 0.983 * | 1.000 * | 0.0037 * |

**Table 2.** Results for the evaluation of the CBCT reconstructions. The dice [†] was evaluated on a segmentation of the large metal object in the phantom. Values marked with an * had significant ($p < 0.05$) changes compared to the refined estimate.

| Algorithm | NGI | SSIM | NRMSE | Dice [†] |
|---|---|---|---|---|
| Coarse estimate | 0.372 * | 0.290 * | 0.7898 * | 0.60 * |
| Refined estimate | 0.525 | 0.360 | 0.5485 | 0.96 |
| Est. + FORCASTER | 0.766 * | 0.677 * | 0.1627 * | 0.99 * |
| Est. + CMA-ES & NGI | 0.944 * | 0.979 * | 0.0295 * | 1.00 * |

Figure 7 shows the sinograms for the sinusoidal trajectory. The calibrated sinograms are the forward projections of the prior CT image. Because this image does not contain the complete object, these two sinograms are missing structures in comparison to the acquired images. One easily spotted difference is in column (b) in the top right quadrant, which is missing a high-contrast object. In the sinogram obtained with estimated starting parameters, i.e., in the bottom row in column (a) and the right image in column (b), there is a distortion visible at the top (column (a)) and left edge (column (b)). This is more pronounced in the sinograms that were calibrated with FORCASTER than the ones calibrated with CMA-ES. This is a projection where the algorithm did not estimate the starting parameters close to the correct ones, and where the calibration did not succeed. This projection is shown in Figure 8 alongside the acquired image and the projection from the correctly calibrated parameters.

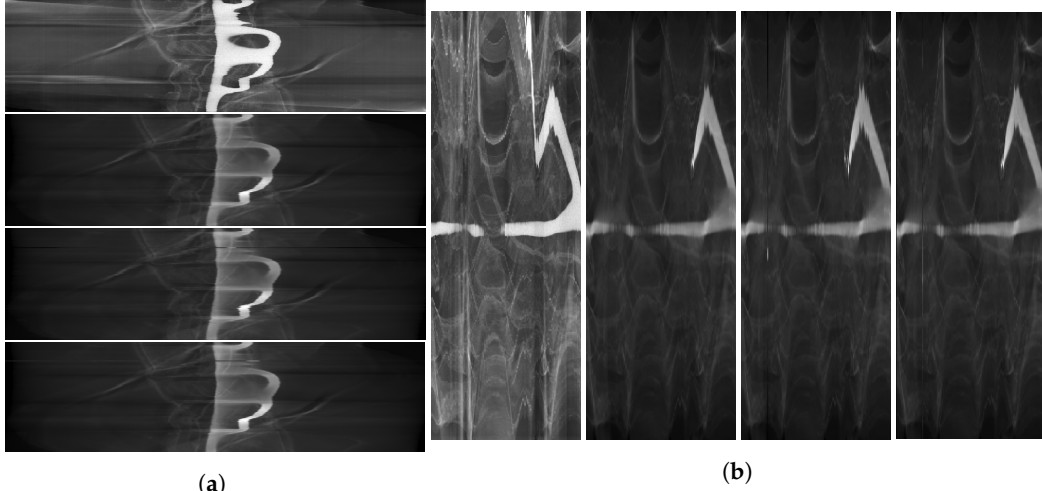

(**a**)                                          (**b**)

**Figure 7.** Two slices out of the sinogram of the sinusoidal trajectory from the acquired data. The FORCASTER calibration used nearly accurate starting parameters. The calibration with refined estimates and FORCASTER, as well as the refined estimates with CMA-ES and NGI objective are shown. The left column (**a**) is ordered from top to bottom as follows: acquired data, simulated projections from the calibration with starting parameters, simulated projections from the FORCASTER calibration with estimated starting parameters, and CMA-ES calibration with the estimated parameters. The right columns (**b**) have the same order, also from left to right.

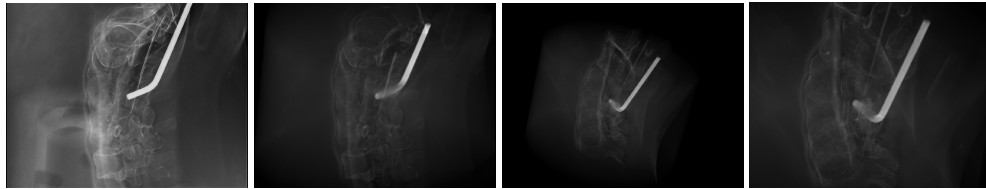

**Figure 8.** Example projections at a point where the parameter estimation fails. From left to right is as follows: acquired projection, simulated projection from the calibration with starting parameters, simulated projection from the calibration with estimated starting parameters and FORCASTER, and the projections from estimated parameters with CMA-ES calibration.

The values obtained from evaluating the metrics are shown in Table 3. Similar to the calibration of the CBCT trajectory, the metrics improved with every step. The final calibration produced lower results than the CBCT calibration, which is in line with the distortions visible in the sinogram. The runtime was much shorter because the sinusoidal trajectory contained only 161 projections in comparison to the 496 contained in the CBCT trajectory.

**Table 3.** Results of the evaluation on the sinusoidal trajectory. Values marked with an * had significant ($p < 0.05$) changes compared to the refined estimate.

| Algorithm | Runtime [hh:mm] | NGI | SSIM | NRMSE |
|---|---|---|---|---|
| Coarse estimate | 00:24 | 0.259 * | 0.885 * | 0.4513 * |
| Refined estimate | 00:27 | 0.662 | 0.992 | 0.0851 |
| Est. + FORCASTER | 00:54 | 0.836* | 0.997 | 0.0369 * |
| Est. + CMA-ES & NGI | 01:50 | 0.955 * | 0.999 * | 0.0106 * |

The results for the continuous arc trajectory are shown in Table 4. Because there was no correct calibration for this trajectory (since there were no starting parameters for the individual projections), the metrics were not evaluated on the simulated projections but were instead evaluated on the acquired images. The metric values were much lower there than for the ones for the two previous trajectories. Still, they significantly improved with each step, and there were no significant differences in the SSIM ($p = 0.5$) and NRMSE ($p = 0.9$) metrics between the two state-of-the-art algorithms.

For comparison, the calibrations for the other two trajectories on the acquired images were evaluated and are included in the results located in the lower part of the table.

**Table 4.** Results of the evaluation on the continuous arc trajectory. Here, the forward projection is compared to the acquired data, and the evaluation results for the other two trajectories are also included if the forward projections from the correct calibration were compared to the acquired data. Values marked with an * had significant ($p < 0.05$) changes compared to the refined estimate.

| Algorithm | Runtime [hh:mm] | NGI | SSIM | NRMSE |
|---|---|---|---|---|
| Coarse estimate | 01:26 | 0.123 * | 0.532 * | 0.7019 * |
| Refined estimate | 01:31 | 0.205 | 0.626 | 0.6359 |
| Est. + FORCASTER | 03:43 | 0.240 * | 0.641 * | 0.6283 * |
| Est. + CMA-ES & NGI | 07:23 | 0.259 * | 0.643 * | 0.6284 * |
| Correct cal. CBCT traj. | 00:35 | 0.294 | 0.377 | 0.7894 |
| Est. + FORCASTER CBCT traj. | 03:28 | 0.286 | 0.392 | 0.7817 |
| Correct cal. sinus traj. | 00:10 | 0.194 | 0.486 | 0.7291 |
| Est. + FORCASTER sinus traj. | 00:37 | 0.183 | 0.481 | 0.7311 |

In Figure 9, the acquired sinogram and the simulated sinogram that used the estimated parameters, which was also calibrated by the CMA-ES with the NGI objective, are shown. Similar to the sinusoidal trajectory, there were some projections where the estimated parameters were wrong. These can be seen in the lower half of column (a), and on the right of column (b). Apart from these few slices, no major misalignment can be seen.

The reconstructed image that used the continuous arc and the calibration is in Figure 10. The image was reconstructed using the FDK algorithm from the astra toolbox without any postprocessing. The images show that the object was reconstructed with just a few artifacts (which came from miscalibration). The results of the metrics are in Table 5. Since there exists no correct calibration and reconstruction of this set of projections, the image was compared with the image obtained by reconstructing the CBCT trajectory. The NGI, SSIM, and NRMSE metrics all increased significantly with each step. There was no significant increase for the Dice value in the reconstruction with refined estimates and the reconstruction after applying the two calibration algorithms. Still, the high Dice score shows that the metal object was reconstructed with a good quality and at the correct position. The SSIM and

NGI metrics were not as high as the results from the calibration of the CBCT trajectory. In addition, the misaligned projections generated a few artifacts.

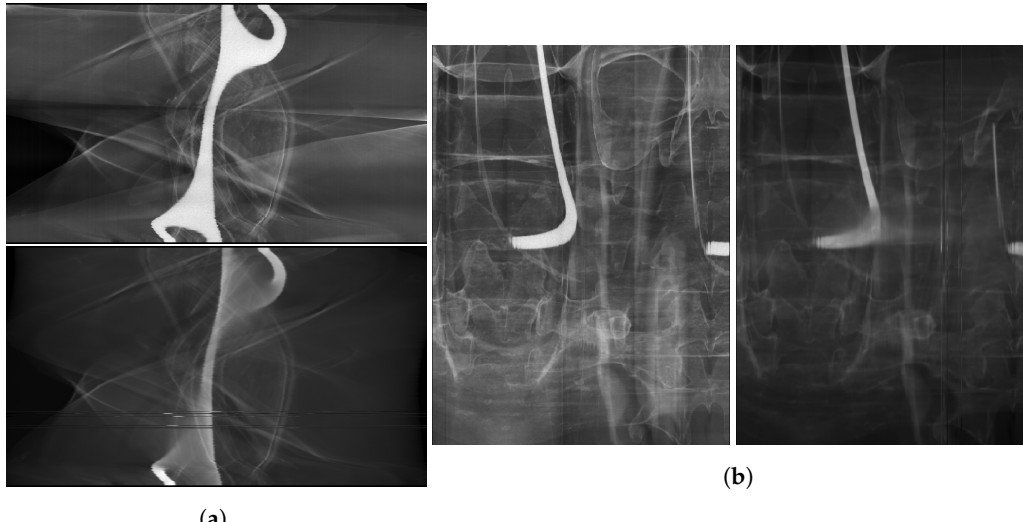

(**a**)

(**b**)

**Figure 9.** Two slices out of the arc trajectory sinogram from the acquired data, as well as from the calibration with refined estimates and CMA-ES with the NGI objective. The left column is (**a**) ordered from top to bottom as follows: acquired data, and the simulated projections from the calibration with estimated starting parameters. The right columns (**b**) have the same order, also from left to right. Since there were no positional data reported by the Artis Zeego System, there was no calibration without estimated parameters conducted.

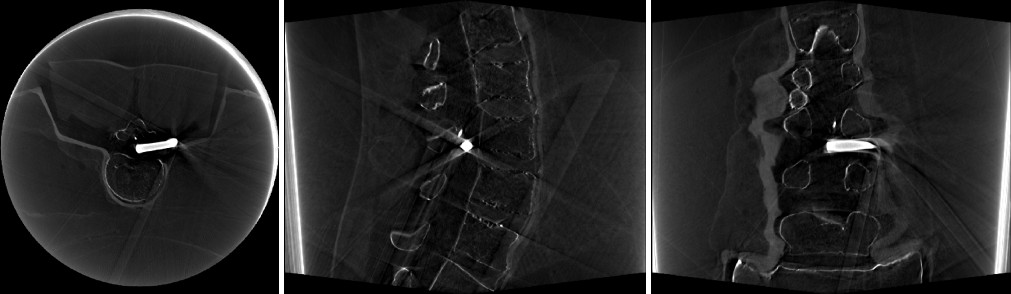

**Figure 10.** Reconstructions of the continuous arc when using the proposed estimator, as well as CMA-ES with the NGI objective and the FDK algorithm from the astra toolbox.

**Table 5.** Metric results for the arc reconstruction. The dice [†] was evaluated on a segmentation of the large metal object in the phantom in comparison to the segmentation of this object in the reconstructed image from the regular CBCT trajectory. Values marked with an * had significant ($p < 0.05$) changes compared to the refined estimate.

| Algorithm | NGI | SSIM | NRMSE | Dice [†] |
|---|---|---|---|---|
| Coarse estimate | 0.353 * | 0.360 * | 0.3567 * | 0.00 * |
| Refined estimate | 0.446 | 0.418 | 0.2379 | 0.82 |
| Est. + FORCASTER | 0.510 * | 0.473 * | 0.2114 * | 0.84 |
| Est. + CMA-ES & NGI | 0.535 * | 0.535 * | 0.1958 * | 0.84 |

## 4. Discussion

This paper describes the FORCAST-EST algorithm, which is able to approximate the initial parameters for the calibration of arbitrary CBCT trajectories. Based on the literature review, this algorithm is the first estimator that uses only online calibration techniques. The results have shown that the estimated parameters are close enough to the correct parameters, such that state-of-the-art algorithms can successfully calibrate the trajectory. The

reconstructed images are of comparable quality to those that have been reconstructed after calibration with close-to-accurate starting positions. The Dice score achieved by both algorithms was very high $\geq 0.99$, which means the metal object in the center was reconstructed correctly. The CBCT calibrations also produced very high SSIM values ($>0.999$), such that they are very close to the reference calibration. These metric values are comparable to those previously reported for the FORCASTER calibration [10]. The estimation, therefore, does not reduce the quality of the reconstructed images. A comparison to the calibration algorithm was developed by Oudah et al. [17], and this was meant to be used in this paper. However, this was not possible since the calibration algorithm uses metrics that require specific objects in the measured phantom, and these were not included in the object scanned for this work.

For the CBCT trajectory, the refined estimate needed about one and a half hours, while together with the CMA-ES algorithm it took five and a half hours for the calibration. As such, the estimation was less than 30% of the total runtime; furthermore, this difference was even higher for the arc calibration, where the estimation only contributed 20%.

Further, in contrast to the approaches by Grzeda et al. or Lemammer et al. [18,19], no modification of the CBCT-Device was necessary to estimate the parameters. If the modification of the C-Arm is possible and allowed, the addition of more sensors might be the preferred option.

Removing the dependency on roughly accurate starting parameters comes with the cost of a higher computation time. In the experiments, there was a steep increase in the time when compared to the calibration that had its starting parameters provided by the CBCT device. Still, every projection was optimized separately from one another, and they also independently used the calibration algorithm. Therefore, parallelization was easily implemented. The initial generation of the grid with projections can also be parallelized, but this was not implemented since the data can be saved and reused for the next time the calibration algorithm is run. This is possible as it only depends on the prior image, and the same grid can be used for the calibration of different trajectories around the object.

One caveat are projections where the acquired image and simulation differ too much. This can happen for projections that are too far out of the plane of the original CT trajectory. The acquired projections contain parts of the object that are not in the prior CT because the phantom is larger than the CT volume. This makes it harder to find enough matching feature pairs. Therefore, the noise of the miss-matched features on one projection might outweigh the low count of the correctly matched points on the correct projection; as such, the estimated parameters would then be incorrect. During the development, it became evident that the parameters like the ratio of Lowe's ratio test had a high impact on how far the projections would be miscalibrated. Further exploration of the different parameter combinations, or other ways through which to filter out wrong matches, might improve this algorithm. The filtering system for matched feature points could also be replaced with the one used by Yang et al. [11] in their calibration algorithm.

More development should be conducted to reduce the runtime. The implementation relies heavily on the CPU, but the graphic card could reduce the needed computation time drastically, since graphic cards are designed for parallel computing.

## 5. Conclusions

In conclusion, this paper presents an algorithm that is able to roughly calibrate a trajectory without initial parameters, and it is accurate enough for state-of-the-art algorithms to further refine the produced image, as demonstrated when using CMA-ES with NGI and the FORCASTER algorithm.

**Author Contributions:** Conceptualization, C.T.; methodology, C.T.; software, C.T.; validation, C.T. and F.G.Z.; resources, C.T.; data curation, C.T.; writing—original draft preparation, C.T.; writing—review and editing, C.T. and F.G.Z.; visualization, C.T.; supervision, F.G.Z.; project administration, F.G.Z.; funding acquisition, F.G.Z. All authors have read and agreed to the published version of the manuscript.

**Funding:** This research project was partially supported by the Research Campus M2OLIE, which was funded by the German Federal Ministry of Education and Research (BMBF) within the Framework "Forschungscampus: Public-private partnership for Innovations" under the funding code 13GW0388A. For the publication fee we acknowledge financial support by Deutsche Forschungsgemeinschaft within the funding programme "Open Access Publikationskosten" as well as by Heidelberg University.

**Institutional Review Board Statement:** Not applicable.

**Informed Consent Statement:** Not applicable.

**Data Availability Statement:** Source code is available at: https://github.com/ChristianToennes/FORCASTER accessed on 1 August 2023.

**Conflicts of Interest:** The authors declare no conflict of interest.

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
