# Peer review of "Feature-Oriented CBCT Self-Calibration Parameter Estimator for Arbitrary Trajectories: FORCAST-EST"

_applsci, doi:10.3390/app13169179_

Round 1

Reviewer 1 Report

The Figure to explain coordinate systems construction should be added.

Pseudocode and/or pipeline chart for algorithm should be added.

Equations for the metrics should be added.

In results NGI, SSIM, NRMSE are used, but there is no information in Metrics about them (abrivations should be added).

Why Figures after the Reference list?

Reference list do not saticify the state of the issue.

According the present edition it’s hard to judge the quality of the presented methods and results. The novelty is not obvious. I advise authors to uncover technical details and then the article should be revised again.

Author Response

Thanks for the review,

I have added a figure for the coordinate systems, a pipeline graph for the algorithm and pseudocode for parts of it.

The metrics have equations, except for the NGI since the equations are quite complex and take up half a page in the original paper, without the needed explanation.

Figures and tables have been moved to the methods/results sections.

I added more references from recent years.

Reviewer 2 Report

It’s a good organized and prepared article which can help radiologists and patients..  

Author Response

Thanks for the review.

Reviewer 3 Report

Dear Authors,

Congratulations on the job you have done and presented in this manuscript. I believe that your job can be of interest for the general reader but for now I cannot recommend publication in such a high ranked journal. You will see that there are major concerns related to the manuscript:

1. there are multiple grammar and spelling errors, English needs to be revised.

2. Part of your methodology is written at the end of the document, conclusion section is missing, there are no cited references in the discussion section, abstract section is missing key element like background of the study or results.....  Please see the attachment.

Extensive language revisions are required

Author Response

Thanks for your review,

I rewrote all the sentences with "we"/"our".

The conclusion has its own section.

All Tables/Figures are now in the methods or results section.

Added background and results to the abstract, but only those from the CBCT reconstruction to keep it short.

Round 2

Reviewer 3 Report

Dear Authors, 

Thank you for taking into consideration my advices and providing the revised version. I have no further comments.

Author Response

Thank you for the review.